# Leveraging mobility data to analyze persistent SARS-CoV-2 mutations and inform targeted genomic surveillance

Riccardo Spott[1]*, Mathias W Pletz[1,2], Carolin Fleischmann-Struzek[1,2], Aurelia Kimmig[1], Christiane Hadlich[3], Matthias Hauert[3], Mara Lohde[1], Mateusz Jundzill[1], Mike Marquet[1], Petra Dickmann[4], Ruben Schüchner[5], Martin Hölzer[6], Denise Kühnert[7,8], Christian Brandt[1,9]

[1]Institute for Infectious Diseases and Infection Control, Jena University Hospital, Jena, Germany; [2]Center for Sepsis Control and Care, Jena University Hospital/ Friedrich Schiller University Jena, Jena, Germany; [3]SMA Development GmbH - epicinsights Agentur für Künstliche Intelligenz und Big Data Analytics, Jena, Germany; [4]Department of Anaesthesiology and Intensive Care, Jena University Hospital, Jena, Germany; [5]Thuringian State Authority for Consumer Protection, Department Health Protection, Bad Langensalza, Germany; [6]Methodology and Research Infrastructure, Genome Competence Center (MF1), Robert Koch Institute, Berlin, Germany; [7]Centre for Artificial Intelligence in Public Health Research, Robert Koch Institute, Berlin, Germany; [8]Transmission, Infection, Diversification and Evolution Group, Max Planck Institute for Geoanthropology, Jena, Germany; [9]Center for Applied Research, InfectoGnostics Research Campus Jena, Jena, Germany

*For correspondence: riccardo.spott@gmail.com

## eLife Assessment

The authors analyze the relationship between human mobility and genomic data of SARS-CoV-2 using mobile phone mobility data and sequence data and present a **solid** proof of concept. This **useful** work was conducted on a fine spatial scale and provides suggestions on how mobility-derived surveillance could be conducted, although these results are mixed. The primary significance of this work is the strong use of large datasets that were highly granular. The authors provide a rigorous study, but with less clear predictive power of mobility to inform transmission patterns.

**Abstract** Given the rapid cross-country spread of SARS-CoV-2 and the resulting difficulty in tracking lineage spread, we investigated the potential of combining mobile service data and fine-granular metadata (such as postal codes and genomic data) to advance integrated genomic surveillance of the pandemic in the federal state of Thuringia, Germany. We sequenced over 6500 SARS-CoV-2 Alpha genomes (B.1.1.7) across 7 months within Thuringia while collecting patients' isolation dates and postal codes. Our dataset is complemented by over 66,000 publicly available German Alpha genomes and mobile service data for Thuringia. We identified the existence and spread of nine persistent mutation variants within the Alpha lineage, seven of which formed separate phylogenetic clusters with different spreading patterns in Thuringia. The remaining two are subclusters. Mobile service data can indicate these clusters' spread and highlight a potential sampling bias, especially of low-prevalence variants. Thereby, mobile service data can be used either retrospectively to assess surveillance coverage and efficiency from already collected data or to actively guide part of a surveillance sampling process to districts where these variants are expected to emerge. The latter concept was successfully implemented as a proof-of-concept for a

mobility-guided sampling strategy in response to the surveillance of Omicron sublineage BQ.1.1. The combination of mobile service data and SARS-CoV-2 surveillance by genome sequencing is a valuable tool for more targeted and responsive surveillance.

## Introduction

On March 11, 2020, the World Health Organization (WHO) classified the SARS-CoV-2 virus (severe acute respiratory syndrome coronavirus 2) as a global pandemic due to its rapid spread and high infection rate (*Zhu et al., 2020*). The airborne virus has since caused significant morbidity and mortality world-wide (https://covid19.who.int/). In an attempt to control its spread, many countries initiated compre-hensive surveillance efforts with molecular techniques such as polymerase chain reaction (PCR) and whole genome sequencing (WGS) (*COVID-19 Genomics UK, 2020*; *Corona-Surveillanceverordnung, 2022*). Consequently, nearly 15.8 million SARS-CoV-2 sequences have been deposited into the 'Global Initiative on Sharing All Influenza Data' database (as of July 21, 2023, GISAID). Many research groups have undertaken studies examining the viral spread by integrating sequencing and epide-miological data to monitor the pandemic and investigate local outbreaks (*Meredith et al., 2020*; *Page et al., 2021*). Most of these local projects are part of national surveillance programs such as the UK's Genomics Consortium (COG-UK) or 'national genomic surveillance' in the USA (*COVID-19 Genomics UK, 2020*; *Lambrou et al., 2021*). In Germany, the 'Coronavirus-Surveillanceverordnung' (CorSurV) enacted by the State Ministry of Health on January 19, 2021, mandated that laboratories with sequencing capabilities process SARS-CoV-2-positive samples, offering financial compensation until April 2023 (*Corona-Surveillanceverordnung, 2022*).

Bioinformatics workflows developed in Germany, such as poreCov (for Oxford Nanopore data) and CoVpipe2 (for Illumina data), reconstruct SARS-CoV-2 consensus genomes from the sequencing data and prepare the results for upload and submission to the Robert Koch Institute (RKI) (*Brandt et al., 2021*; *Lataretu et al., 2024*). As the German government's public health and biomedical research institute responsible for disease control and prevention, the RKI collected the genomes via the German Electronic Sequence Data Hub (DESH) and integrated them with additional epidemio-logical information to provide an up-to-date overview of the ongoing viral spread. For keeping track of the rapid SARS-CoV-2 evolution, PANGO (Phylogenetic Assignment of Named Global Outbreak) provides a standard naming convention based on unique mutation profiles and further criteria, resulting in the classification of over 3660 lineages (as of August 2023) (*Rambaut et al., 2020*; *PANGO, 2023*). Additionally, the WHO classified important viral lineages as 'Variants of Concern' (VOC), 'Variants of Interest' (VOI), or 'Variants under Monitoring' (VUM), using Greek designations in the past (e.g. 'Alpha' [Pango lineage main designation B.1.1.7] or 'Omicron' [Pango lineage main designation B.1.1.529]). Further, the WHO also de-escalated former VOCs to reflect the current SARS-CoV-2 variant landscape better. The first defined VOC (now de-escalated), the Alpha lineage, rapidly replaced almost all previously circulating lineages globally by the end of 2020 until the VOC Delta (main lineage B.1.617.2) replaced it in mid-2021 (*Washington et al., 2021*; *Walker et al., 2021*; *Michaelsen et al., 2022*).

To predict or monitor the rapid viral spread throughout regions, various data types, like travel data, passenger volumes, or passive wastewater monitoring, were examined previously (*Alpert et al., 2021*; *O'Toole et al., 2021*; *Li et al., 2022*). Furthermore, different studies explored mobility data with genomic data to retrace the origin and spatial expanse of Alpha or utilized geolocation data to model the spread in metropolitan areas to recreate case trajectories and the impact of mobility restric-tions (*Kraemer et al., 2021*; *Chang et al., 2021*). Mobility data was also used in Germany during the pandemic, revealing that lockdowns leave distant parts of the country less connected due to the sharp decline in long-distance travel (*Schlosser et al., 2020*). These studies focused on analyzing residential movement and contact tracing to evaluate and inform health policies but were not applied to active molecular surveillance.

Here, we investigated whether mobile service data and fine-granular metadata (such as postal codes and genomic data) can help predict the spread of the Alpha lineage or guide the sampling for more targeted genomic surveillance with a focus on the German federal state of Thuringia.

## Results and discussion

### The Alpha lineage spread rapidly through Thuringia, showing a pattern similar to its nationwide spread

Thuringia is a rural federal state in central Germany with a population of 2.1 million and no major airports (overview of Thuringia's population density in *Figure 1—figure supplement 2*). We investigated if the spread of the Alpha lineage of SARS-CoV-2 behaved differently compared to the whole of Germany. To understand its spread, we used 289,487 public SARS-CoV-2 genomes from Germany (excluding Thuringia; including 137,024 Alpha genomes) and 7394 genomes from our own sequencing data for Thuringia (including 6307 Alpha genomes) to track Alpha's occurrence from December 2020 to August 2021 (see *Figure 1*, *Figure 1—figure supplement 1*, and *Figure 1— source data 1*; *Figure 1—source data 2*; for details, see Methods section 'Alpha spread datasets'). For Thuringia, district-level data (full postal code) per genome were available, whereas, for Germany, only postal code data of the sending laboratories (referred to as 'primary diagnostic laboratory' by the RKI where the SARS-CoV-2-positive sample was detected) and sequencing laboratories were publicly available.

In late December 2020, six federal states in Germany (from here on called states) reported the first cases of the Alpha variant. Although sequencing was initially low, it gradually increased in the following month. However, the Corona-Surveillance regulation was passed at the end of January 2021, leading to a rapid increase in sampling and sequencing by February since sequencing costs could be reimbursed. Even though Thuringia sequenced a similar amount of SARS-CoV-2 samples compared to other German states (as shown in *Figure 1*), the proportion of the Alpha variant to other lineages was relatively low. However, the proportion of Alpha increased heavily in February.

By March, Alpha had spread to nearly all states and districts (districts are similar to counties or provinces) in Germany (median: 76.47% Alpha samples among a federal states total sequenced samples compared to 36.03% in February, excluding Thuringia) and Thuringia (median: 85.29 %, up from 50.00% in February). So, there was no noticeable difference in the Alpha proportions between Germany and Thuringia after February. During the summer of June and July 2021, sequencing declined in Germany (including Thuringia; *Figure 1—figure supplement 1*) due to the decrease in overall daily cases, as reported by *Meintrup et al., 2022*; *Oh et al., 2022*.

In summary, the spread of the Alpha lineage in Thuringia lagged roughly 2 weeks behind the general spread of other German federal states but showed similar proportions. This suggests that Thuringia experienced a delay in the initial arrival of Alpha. However, we did not observe any difference in the overall spread afterward. Thuringia was among the first states to adopt new containment measures, including contact limitations, closure of retail shops, and prohibition of tourist journeys (December 14, 2020). Jena, a city in Thuringia, was also the first German city to implement mandatory public masking in March 2020 (*Pletz et al., 2023*). Contacts were further restricted on January 9th, and people were urged to restrict their movement radius to 15 km, which might explain the delay besides the absence of major airports nearby.

All Thuringian genomes were evenly distributed between other German samples in the phylogenetic time tree (see *Figure 1—figure supplement 3*). However, due to its rapid spread from February onward, it is difficult to accurately track how the Alpha lineage specifically expanded (point of entries, exact origins, etc.). Consequently, we investigated whether 'sublineages' might be identifiable and trackable to address this.

### Monitoring of Alpha subclusters in Thuringia reveals temporally and regionally restricted distribution patterns

To identify possible clusters among the Alpha lineage spreading in Thuringia, we called each Alpha genome's mutations via Nextclade by analyzing them using poreCov (*Brandt et al., 2021*; *Aksamentov et al., 2021*). We identified nine clusters out of 70,429 Alpha genomes, based on their mutation profile, time period, and phylogenetic distance (from here on called Alpha subclusters; for details, see Methods 'Subcluster identification'). All subclusters, their time period, and sample size in Thuringia are summarized in *Table 1*. An overview of each subcluster (phylogenetic time tree, location, and period) is also provided here as interactive views (see Methods 'Subcluster identification'). Note that our subcluster definition is similar to the definition of a sublineage. However, PANGO

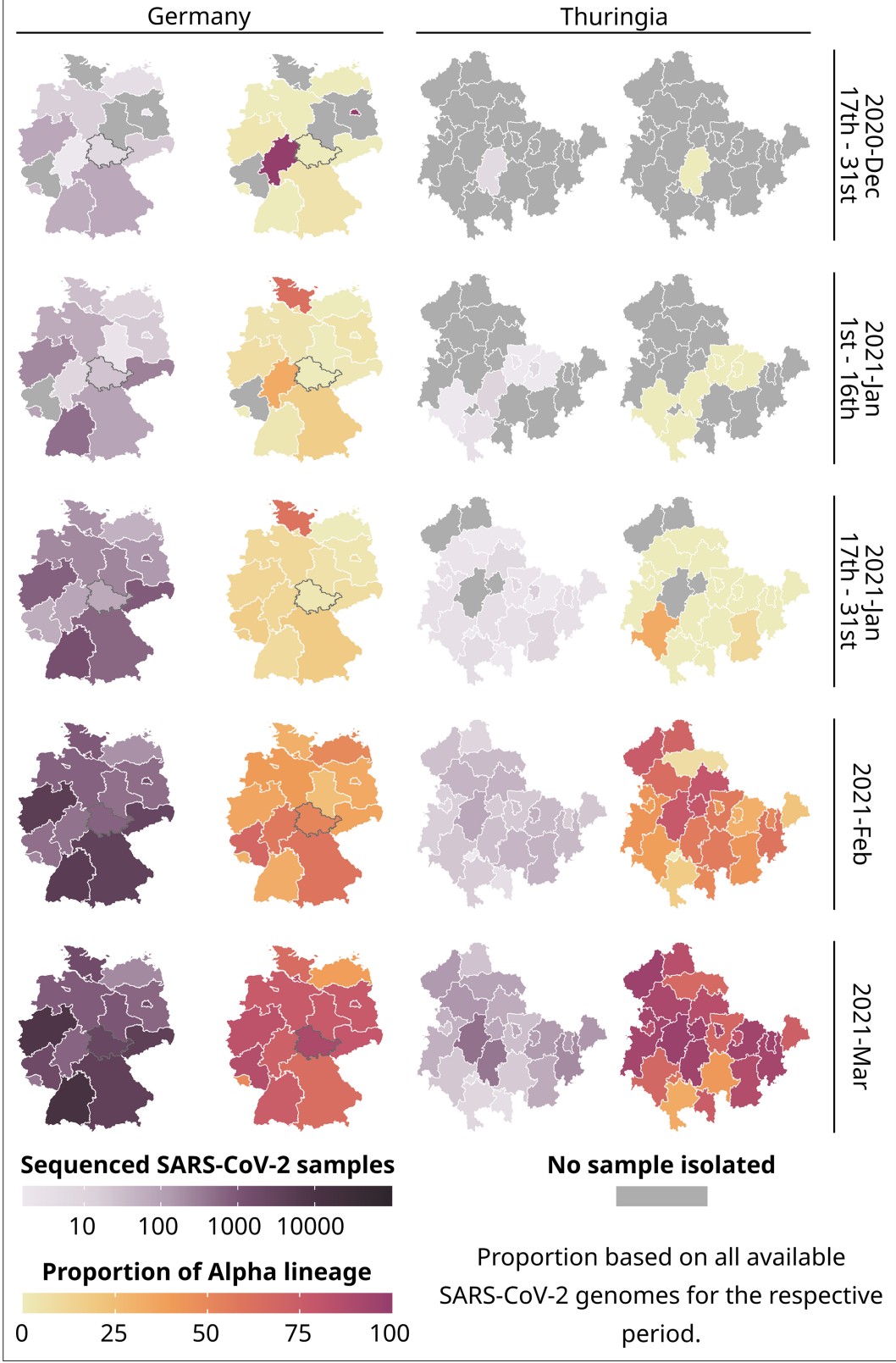

**Figure 1.** Total number of all sequenced SARS-CoV-2 samples (purple) and the proportion of the Alpha lineage for all sequenced samples (yellow-red) for each state of Germany and each district of Thuringia. 289,487 publicly available German SARS-CoV-2 genomes and their metadata were used for the general German maps, excluding data from Thuringia. For Thuringia, we always used 7394 genomes and their metadata from our database for the

*Figure 1 continued on next page*

*Figure 1 continued*

German and Thuringian maps. Please note that for all states except Thuringia, we used the postal code of the sending laboratory as a proxy for the geographical location of a sample. A gray border on the maps of Germany highlights Thuringia.

The online version of this article includes the following source data, source code, and figure supplement(s) for figure 1:

**Figure supplement 1.** Total number of all sequenced SARS-CoV-2 samples (purple) and the proportion of the Alpha lineage for all sequenced samples (yellow-red) for each state of Germany and each district of Thuringia throughout the whole observation period.

**Figure supplement 2.** Total population (**a**) and population density per km$^2$ (**b**) for each Thuringian district as stated for the December 31, 2020.

**Figure supplement 3.** Phylogenetic time tree of the Alpha lineage.

**Source code 1.** R-script to generate *Figure 1*, *Figure 1—figure supplement 1*.

**Source data 1.** Total number of reported SARS-CoV-2 samples and number of reported Alpha lineage samples for each German federal state per month.

**Source data 2.** Total number of reported SARS-CoV-2 samples and number of reported Alpha lineage samples for each Thuringian district per month.

sublineages were rarely defined during the Alpha wave (PANGO designation: Q.1 to Q.8; compared to the Delta and Omicron waves).

Eight of these subclusters are based around a specific spike protein mutation, while the other contains a mutation within the ORF1b region. The subcluster 7.1 'S:N185D' branched out from the subcluster 7 'ORF1b:A520V' and subcluster 6.1 'S:V90F' is a branch of subcluster 6 'S:S939F' (see *Table 1*). These two branched subclusters still carry the specific mutation of their originating subcluster. The subclusters 3, 4, and 5 were observable between two and three months, and the other subclusters

**Table 1.** Overview of nine Alpha subclusters in Thuringia, their sample count, their time period, and their specific mutations that are shared across all members of the subcluster (excluding characteristic Alpha mutations that are shared across all subclusters). The mutation used to define the subcluster is highlighted in bold.

| Designation | Mutations | Number of samples | Time period | Remarks |
|---|---|---|---|---|
| 1 | **S:H49Y,** ORF1a:I841V | 44 | Feb-May 2021 | S:H49Y eases cell entry in S-pseudotyped lentiviral system (*Ozono et al., 2021*) |
| 2 | **S:N354K** | 63 | Feb-May 2021 | S:N354K slightly impaired mAb h11B11 (*Du et al., 2021*) |
| 3 | **S:G496S,** ORF1a:E1013K | 12 | Mar-May 2021 | S:G496S: compromises BA.1 replication fitness and changed mAb sensitivities, reduces ACE2 binding affinity, and increases immune evasion (*Liang et al., 2022*; *Kimura et al., 2022*; *Asif et al., 2022*) |
| 4 | **S:N703D,** ORF1a:D1228G, ORF1a:A2123V | 51 | Mar-May 2021 | – |
| 5 | **S:T716V,** N:G204P, ORF1a:D1600N | 22 | Apr-May 2021 | – |
| 6 | **S:S939F** | 206 | Feb-May 2021 | S:S939F: modulates T-cell propensity (*Donzelli et al., 2022*) |
| 6.1[†] | **S:V90F,** S:S939F | 55 | Feb-May 2021 | – |
| 7 | **ORF1b:A520V** | 811 | Feb-Jun 2021* | – |
| 7.1 [‡] | **S:N185D,** ORF1b:A520V, ORF1b:L1504F | 40 | Feb-May 2021 | – |

*Only one sample for June.
[†]Branch from subcluster 6.
[‡]Branch from subcluster 7.

over at least 4 months. To investigate these subclusters' regional spread, each sample was mapped to its Thuringian district based on the resident's postal code from which it was isolated. We then sorted the samples according to their subclusters and visualized them throughout the subcluster's observed period. The spread of two representative subclusters is exemplary visualized in *Figure 2a*, and all the subclusters are available via *Figure 2—figure supplement 1* and their data in *Figure 2—source data 1*. Additionally, all subclusters and their metadata are also available via Microreact (see Methods 'Subcluster identification').

In Thuringia, the seven main mutation variant clusters were initially identified from distinct districts. Subclusters 6.1 and 7.1 subsequently emerged from the same districts as their parent clusters (6 and 7) after 12 and 13 days, respectively. The subclusters mainly spread regionally confined and not across all of Thuringia (see *Figure 2a*, *Figure 2—figure supplement 1*) but were also identified in other states of Germany (see 'https://microreact.org/'-project). For example, the 'S:S939F' subcluster spread across 15 states, with the first samples being isolated outside of Thuringia. The eight Spike-mutation subclusters had expanded between 4 and 12 of the 23 Thuringian districts within the observable time period of each subcluster. They expanded by one to six districts per month, with a greater expansion accompanied mainly by a larger increase in the subcluster sample number. In contrast, the ORF1b-variant even comprised 21 districts and expanded between 2 and 7 districts per month (see *Figure 2a*). Most of each subcluster's samples were identified in their region of first occurrence, and no additional samples were found after the given periods.

Several limitations need to be considered. The identified subclusters may have multiple origins or may not originate from Thuringia. Due to the lack of precise zip codes (publicly available German genomes are limited to postal codes of sending and sequencing laboratories), monitoring the subclusters in other states on a district level was impossible. Nevertheless, we could follow how the subclusters developed in Thuringia, even if multiple origins may have affected the overall speed or length of each subcluster's occurrence.

Our surveillance sampling heavily relies on various institutions and partners, and only a portion of the provided samples can be sequenced (see 'Sampling' in Methods). For example, the spread of subcluster 'S:S939F' revealed two districts in April where no respective samples were found (*Figure 2a*) despite being surrounded by districts with 'S:S939F'-samples present. This could be due to the lack of samples sent to sequencing from those regions or the low prevalence. We, therefore, investigated if mobile service data of residents, in addition to molecular surveillance, might be utilized to counteract this issue.

## Mobile service data indicates Alpha subcluster spread and sampling bias

With the aim to predict the subcluster spread and, thereby, reduce surveillance-based sampling bias, we utilized anonymized mobile service data from T-Systems International GmbH. Around 200 million trips were used to determine the number of daily trips between the Thuringian districts. We then combined this information with our fine-granular genomic data to specify each district's monthly proportion of inbound mobility from subcluster-affiliated districts (see Methods 'Mobile service data'). The results are visualized in *Figure 2b* (complete overview in *Figure 2—figure supplement 2*; data provided in *Figure 2—source data 2*).

The mobile service data-based assumption of a subcluster's spread aligned well with the subsequent regional coverage of fast-spreading, highly prevalent subclusters, such as subcluster 7, which covered 811 samples (see *Figure 2*). In contrast, the assumed spread for the low-prevalence subclusters did not correspond well with the actual occurrence. Yet, adding mobile service data resulted in three different 'types' of districts (see *Figure 2b*, annotated districts). Type 1 included districts with high inbound mobility from areas with an identified variant, where the variant was eventually found afterward, while Type 2 included districts with high inbound mobility from areas with an identified variant, where the variant was never identified. Type 3 included districts not directly connected to a district with an identified variant, but a variant was eventually identified while they border Type 2 district(s).

Our previous analysis of the subclusters' spreading pattern across the districts, based solely on identified variants, indicated putative missed identifications in some districts due to the seemingly illogical spread to districts without a connection to others (*Figure 2a*, subcluster 'S:S939F'). The

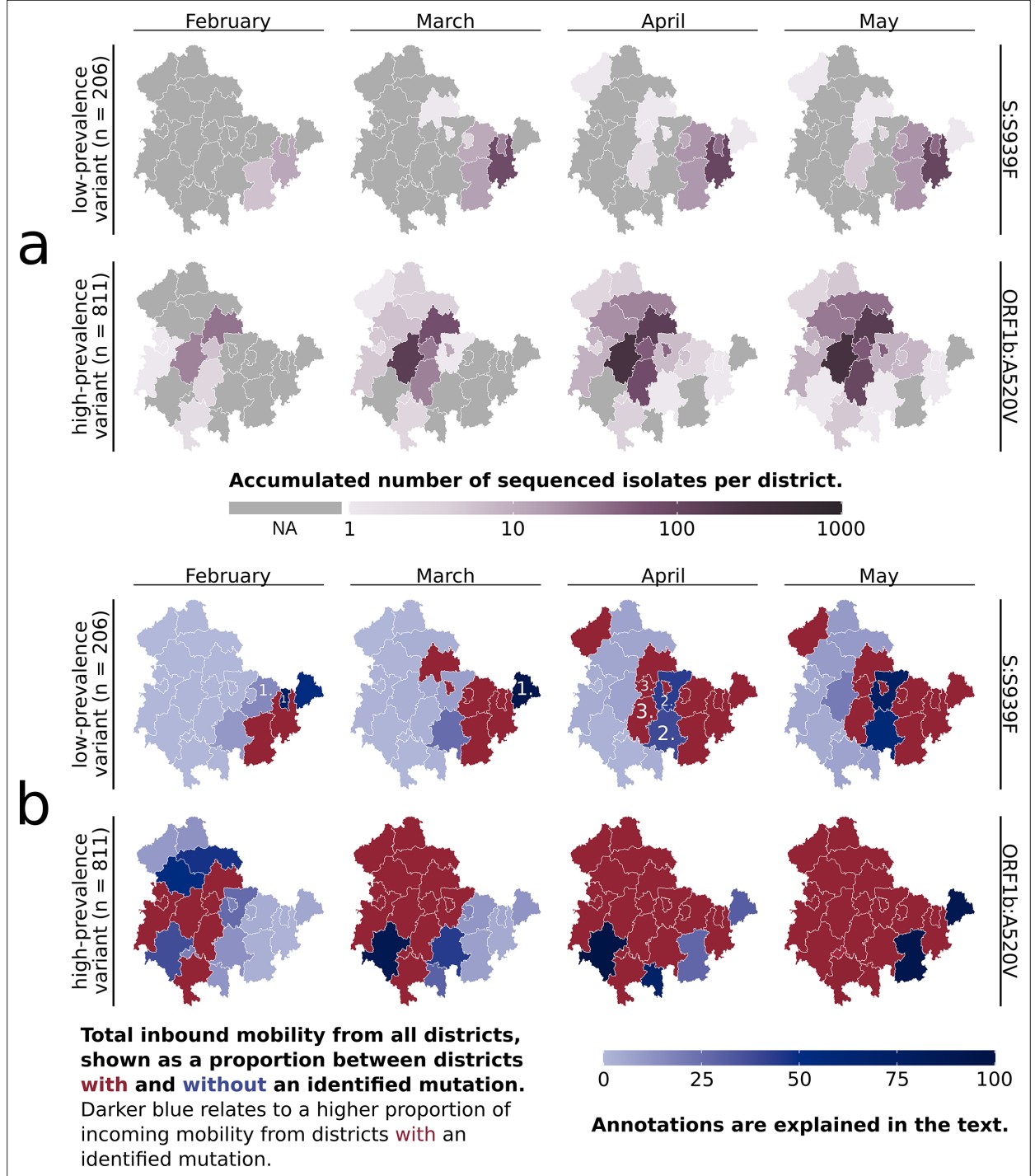

**Figure 2.** Overview of the subclusters 'S:S939F' and 'ORF1b:A520V' in Thuringian districts. (**a**) Accumulated number of sequenced samples for each subcluster per district and per month. (**b**) Combined visualization of each district's 'inbound mobility' from other districts (color intensity) and the occurrence of a subcluster sample (red = sample found, blue = no sample found). The inbound mobility of each district (blue color intensity) is shown as a proportion of incoming mobility from other districts with or without an identified sample. The darker the blue color of a district, the higher the proportion of inbound mobility from other districts with an identified subcluster sample (red districts). The light blue color describes that most of the inbound mobility of a district comes from other districts without an identified subcluster sample (blue districts). Numbers refer to district types 1, 2, and 3, as further defined in the main text. The last month of subcluster 'ORF1b:A520V' is not visualized, as affected districts were unchanged.

The online version of this article includes the following source data, source code, and figure supplement(s) for figure 2:

**Source code 1.** R-script to generate *Figure 2*, *Figure 2—figure supplements 1 and 2*.

*Figure 2 continued on next page*

Figure 2 continued

**Source data 1.** Accumulated sample count for each Thuringian district per Alpha subcluster and month.

**Source data 2.** Total number of incoming trips and numbers of trips coming from all cluster-affiliated districts to each Thuringian district per Alpha subcluster and per month.

**Figure supplement 1.** Accumulated number of sequenced samples for each Alpha lineage subcluster per district and per month.

**Figure supplement 2.** Combined visualization of each district's 'inbound mobility' from other districts (color intensity) and the occurrence of a subcluster sample (red = sample found, blue = no sample found) per subcluster.

inclusion of mobile service data revealed some of these districts to be Type 2 districts. This suggests that the specific variant should be identifiable within these districts due to the observed high incoming mobility from districts with identified variants. Type 2 districts were mainly observed for subclusters with low prevalence and, consequently, low sample counts, which are usually more difficult to monitor. For example, we assumed missing identifications in some districts of subclusters 1, 2, and 3, which, through the mobile service data, are now partially identified as Type 2 districts. However, due to their low prevalence, it is also possible that these subclusters have not spread to the indicated districts. Despite analyzing the mobile service data of districts from other federal states than Thuringia, we could not apply them, as the lack of precise location data for samples outside of Thuringia prevented the correct calculation of the incoming mobility. Based on the nine observable clusters, we concluded that mobile service data is a good prediction marker for the spread of high-prevalence variants but, more importantly, a good indication of districts that should have an identified low-prevalence variant. Next, we investigated if mobile service data can improve active surveillance via guiding sample collection for genomic sequencing.

## Proof of principle: mobile service data-guided sampling for genomic surveillance for Omicron BQ.1.1

Based on our previous findings, we implemented the 'mobility-guided' sampling approach under real pandemic circumstances over 1 month in addition to our active surveillance.

As the subject of investigation, we searched for a newly emerging (based on global news reports) and ideally low prevalent SARS-CoV-2 lineage in Thuringia.

Among the various emerging Omicron sublineages during that time, sublineage BQ.1.1 fulfilled the defined criteria. First isolated in a northwestern Thuringian community with around 20,000 inhabitants on October 5, 2022, we identified this particular sublineage on October 14, 2022, among a routine batch of 42 samples. BQ.1.1 was a low-prevalence sublineage that was identified worldwide (https://outbreak.info/situation-reports?pango=BQ.1.1).

Following its first Thuringian identification, we utilized the latest available dataset of the past 2 years of mobile service data (October 2020 and June 2021) to investigate the residential movements for the community of first detection. Considering the highest incoming mobility from both datasets, we identified 18 communities with high (>10,000), 34 with medium (2001–10,000), and 82 with low (30–2000) number of incoming one-way trips from the originating community (purple triangles in *Figure 3a*). As a result, we specifically requested all the available samples from the eight communities with the highest incoming mobility. Still, we were restricted to the submission of third parties over whom we had no influence. This led to the inclusion of the following eight communities with the most residential movement from the originating community: four in central and three in NW of Thuringia, one in NW-neighboring state Saxony-Anhalt. The samples requested from central Thuringia were also due to their geographical arrangement as a 'belt' in central Thuringia, linking three major cities (see *Figure 1—figure supplement 2*). Subsequently, we collected 19 additional samples (isolated between October 17 and October 25, 2022; see 'Guided Sampling' for October 2022, *Figure 3a*) besides the randomized sampling strategy. Thus, the sampling depth was increased in communities with high incoming mobility from the first origin.

As part of the general Thuringian surveillance, we collected 132 samples for October (covering dates between the 5th and 31st) and 69 samples in November (covering dates between the 1st and 25th; see *Figure 3b and c*). Randomized sampling was not influenced or adjusted based on the mobility-guided sample collection. Thus, it also contains samples from communities with a mobility link toward the first occurrence of BQ.1.1, as they were part of the regular random collection (see gray

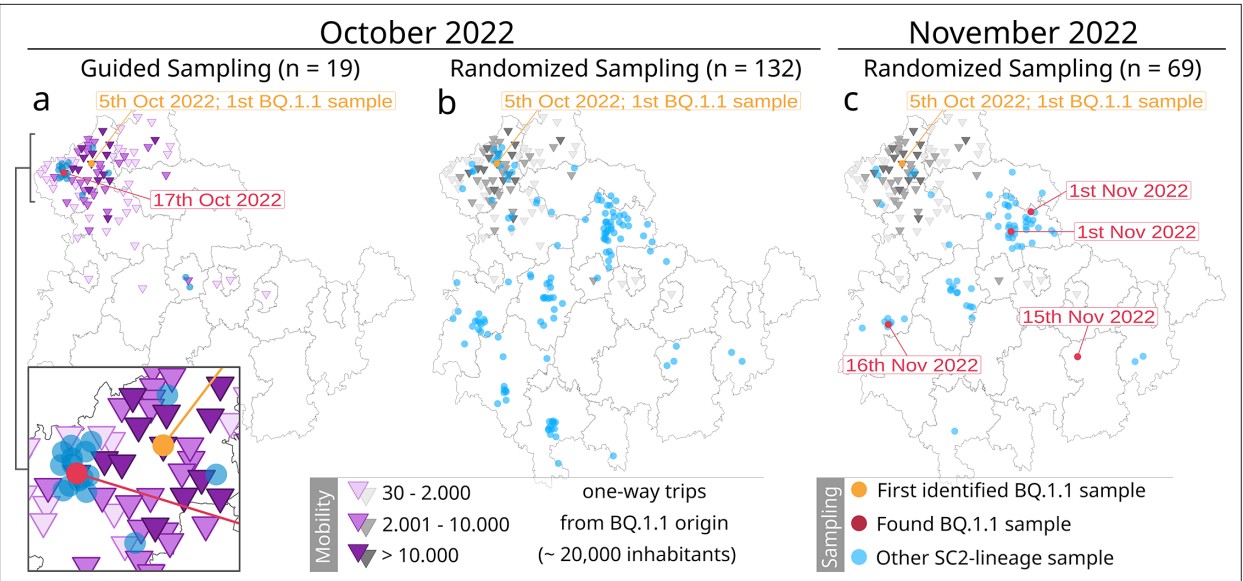

**Figure 3.** Overview of the mobility-guided sampling of the Omicron sublineage BQ.1.1 in Thuringia (**a**) compared to the default randomized sampling (surveillance) in October (**b**) and November 2022 (**c**). The randomized surveillance results in November 2022 (**c**) have been added to highlight the spreading progress of BQ.1.1. Dots reflect the location of each sample (based on residents' zip codes). Orange dot: first identified BQ.1.1 sample; red dot: additionally identified BQ.1.1 sample; blue dot: another SARS-CoV-2 lineage. Purple triangles represent the number of one-way trips a community received from the community of the first BQ.1.1 occurrence (orange dot) based on mobility data from October 2020 and June 2021. The same mobility data from mobility-guided sampling (**a**) were added in grayscales to the randomized surveillance (**b and c**) as a visual reference only. Sampling dots are slightly scattered to improve visibility.

The online version of this article includes the following source data for figure 3:

**Source code 1.** R-script to generate *Figure 3*.

**Source data 1.** Overview of all Thuringian samples collected for the mobility-guided pilot experiment between October 5 and November 25, 2022.

**Source data 2.** Mobility data and sampling counts of communities sampled for the mobility-guided pilot experiment between October 5 and November 25, 2022.

triangles in *Figure 3b*). A complete overview of all samples is provided in *Figure 3—source data 1*. The mobility datasets from October 2020 and June 2021 for all sampled communities are provided in *Figure 3—source data 2*.

Among the 19 samples specifically collected based on mobile service data, we identified one additional sample of the specific Omicron sublineage BQ.1.1 in a community with high incoming mobility (n=14, number of trips = 37,499) with a distance of approximately 16 km between both towns. Our randomly sampled routine surveillance strategy did not detect another sample during the same period. This was despite a seven times higher overall sample rate, which included 31 samples from communities with an identified incoming mobility from the community of the first occurrence (October 2022, *Figure 3b*). Only in the 1-month follow-up were four other samples identified across Thuringia through routine surveillance (November 2022, *Figure 3c*).

During our attempt to implement the mobility-guided sampling approach in real-time during the pandemic, we encountered three distinct limitations, some of which are commonly observed in surveillance practices. The guided sampling depended on the individual sample submitting institutions, affecting the availability of suitable samples, especially for the communities of interest. By choosing a newly emerging Omicron sublineage for our experiment, spread and, therefore, suitability were uncertain. In our case, BQ.1.1's prevalence in Thuringia was even lower than expected, and it also remained rare in subsequent months, with only 42 samples found until June 2023, 8 months after the first occurrence in Thuringia. Due to the short preparation time, only mobile service data from the past 2 years and no current data were available. Nevertheless, the available datasets still reflect pandemic movement behavior since the pandemic has been ongoing for 2 years.

In summary, increasing the sampling depth in the suspected regions successfully identified the specified lineage using only a fraction of the samples from the randomized sampling. Conversely,

randomized surveillance, the 'gold standard' acting as our negative control, did not identify additional samples with similar sampling depths in regions with no or low incoming mobility or even in high mobility regions with less sampling depth. Implementing such an approach effectively under pandemic conditions poses difficult challenges due to the fluctuating sampling sizes. Although the finding of the sample may have been coincidental, our proof-of-concept demonstrated how we can leverage the potential of mobile service data for targeted surveillance sampling.

## Conclusion

During the SARS-CoV-2 pandemic, diverse data sources like travel, wastewater, and mobility data have been employed in surveillance and transmission tracking (*Alpert et al., 2021*; *O'Toole et al., 2021*; *Li et al., 2022*; *Kraemer et al., 2021*; *Chang et al., 2021*; *Schlosser et al., 2020*). In the present study, we analyzed over 296,800 German SARS-CoV-2 genomes to examine whether mobile service data can predict the spatial distribution of the Alpha lineage in the German state of Thuringia and how they potentially benefit pandemic surveillance.

A plausible explanation for the delayed spread of the Alpha lineage in Thuringia is the lack of major transport hubs, as Alpha first occurred in federal states with such hubs. Previous studies have already highlighted the impact of major transportation hubs in the spread of Sars-CoV-2 (*Alpert et al., 2021*; *Tegally et al., 2023*). However, its impact on the total distribution is limited, and the spread was ultimately comparable between Germany and Thuringia. While our findings on mobile service data may also apply to Germany, we could not verify this because the limited location data of publicly available German genomes prevented in-depth investigations outside of Thuringia. Thus, precise sampling location data are crucial to utilize mobile service data in genomic surveillance, but privacy regulations may restrict access to this data. Shortly after Alpha's emergence, mutation variants formed like the known sublineages Q.1 to Q.8 and the Thuringian subclusters identified by us. This reflects the ongoing evolution during active circulation and indicates an even greater sublineage diversity, which has not been surveyed as closely as in the subsequent Delta and Omicron waves. By monitoring the nine Thuringian subclusters, rather than focusing solely on the parental lineage B.1.1.7, we were again able to effectively track transmissions and gain a comprehensive understanding of the regional spread. So, it underscores the importance of sequencing in pandemic surveillance to explore such genomic changes and, thereby, keep track of the transmission chains and potential outbreaks.

Mobile service data can support such surveillance in different ways. Previous studies examined the capabilities of mobility data in the context of, e.g., case trajectories, but retrospectively applied to already collected data, it can be used to examine surveillance sampling coverage and possible sampling bias. We exemplified this approach with the Alpha lineage, where mobile service data indicated a putative sampling bias and partially predicted the spread of our Thuringian subclusters.

Another approach is actively guiding the sampling process through mobile service data, which we demonstrated with our proof of principle focusing on the Omicron-lineage BQ.1.1 as a real-life example. This approach could allow for a flexible allocation of surveillance resources, enabling adaptation to specific circumstances and increasing sampling depth in regions where a variant is anticipated. By incorporating guided sampling, much fewer resources may be needed for unguided or random sampling, thereby reducing overall surveillance costs.

Additionally, while this approach is particularly useful for identifying low-prevalence variants, it is not limited to such variants. Still, it can provide a guided, more cost-efficient, low-sampling alternative to general randomized surveillance that can also be applied to other viruses or lineages. For this purpose, pre-generated mobility networks automatically tailored to each state's unique infrastructure and population dynamics could provide better-targeted sampling guidance rather than simple geographical proximity. However, the feasibility depends on the availability and cost of such mobile service data. Alternatively, financial resources could also be invested directly in increasing sampling capacity and coverage, which ultimately depends on individual factors of the respective surveillance. Mobile service data can also be used with other surveillance approaches and elements. For example, wastewater surveillance can give further indications to supplement guided sampling. At the same time, passenger data offers additional insights into traffic hubs as sources of regional movement.

## Methods

### Sampling

Starting mid-2020, we initially sequenced hospital-intern samples, transitioning by January 2021 to approximately 43 PCR-positive samples per week: 20 from the hospital's microbiology department and 23 randomly sourced by the Thuringian State Authority for Consumer Protection ('Thüringer Landesamt für Verbraucherschutz' [TLV]).

Until June 2023, our institute sequenced 3770 SARS-CoV-2 samples, and SYNLAB Holding Deutschland GmbH, Bioscientia Healthcare GmbH, and DIANOVIS GmbH provided additional 7800 Thuringian SARS-CoV-2 genomes and their metadata.

### Sample preparation and sequencing

RNA isolation used the ZymoResearch 'Quick-RNA Viral Kit' (Zymo Research Europe GmbH, Germany, Product-ID: R1035), according to the manufacturer's instructions with 100 µl patient sample input and a centrifuge speed of 16,000×*g*.

The viral RNA underwent a reverse transcriptase (RT)-PCR followed by a multiplex-PCR using the ARTIC V1200 primer set, according to Freed and Silander's SARS-CoV-2 sequencing protocol (version 4, updating to version 5 by March 2021) (*Freed and Silander, 2021*). Subsequent DNA quantification utilized the Qubit dsDNA HS assay (Invitrogen, USA).

From the amplified DNA, a sequencing library was prepared using the Nanopore SQK-LSK109 and SQK-RBK004 kits (Oxford Nanopore Technologies, Oxford, UK), sequenced for a maximum of 72 hr utilizing an Oxford Nanopore MinION Mk1b sequencer with R.9-flowcells and the MinKNOW software (versions MKE_1013_v1_revBC_11Apr2016 to MKE_1013_v1_revBR_11Apr2016 in the respective period), and analyzed with the software pipeline poreCov (versions 0.3.5–0.11.7; including basecalling, demultiplexing, adapter removal, quality filtering, and genome alignment) to reconstruct consensus genomes (*Brandt et al., 2021*).

Sequencing data and the respective metadata (e.g. isolation date, sending laboratory details) were submitted to the RKI through DESH. We also collected the postal code of the isolation location or at least of the sending local health authority, storing all data additionally in a local database (*Jundzill et al., 2023*). Due to data protection, such data is limited on the RKI's public GitHub repository (https://github.com/robert-koch-institut/SARS-CoV-2-Sequenzdaten_aus_Deutschland; *Robert Koch-Institut, 2025*), providing instead postal codes of the sequencing and sending laboratories.

### Alpha spread datasets

From our local database, we extracted 8397 samples with isolation dates before October 1, 2021. After adding federal state and district information, 993 entries with non-Thuringian locations were excluded. Further, 10 entries with unspecific isolation dates were excluded, yielding 7394 samples (including 6307 Alpha genomes [lineages B.1.1.7 and Q.1 to Q.8]).

The publicly available RKI SARS-CoV-2 dataset was downloaded, containing 1,091,655 genomes with the respective metadata (October 17, 2022; Zenodo-version October 16, 2022) (*Koch-Institut, 2022*). 789,405 entries, isolated after September 2021, and 59 entries without 'sending laboratory' information were removed. For the resulting 302,191 entries, location information (location, federal state, district, longitude, latitude) were added based on the sending laboratory postal code. Five entries with a non-existing postal code and all 12,704 Thuringian samples were removed from the dataset, resulting in 289,487 samples (including 137,024 Alpha genomes). We investigated only Alpha lineage samples collected from September 2020 onward, after the first official reports of the Alpha lineage (*Washington et al., 2021*).

Analyzing both datasets, we calculated the monthly proportion of Alpha lineage samples in Thuringia and Germany per state/district, dividing December 20 and January 2021 into first and second halves.

### Subcluster identification

Using a total of 70,429 German and Thuringian Alpha genomes, a phylogenetic time tree was created (see *Supplementary file 2*, *Supplementary file 3*, *Supplementary file 4*, *Supplementary file 5*, *Supplementary file 6*, *Supplementary file 7*, *Supplementary file 8* and *Figure 1—figure supplement 3*). We determined the frequency of all non-Alpha-specific mutations among the 6522 Thuringian

Alpha genomes. We then manually screened for mutations present in at least 20 genomes with a small phylogenetic distance and a time occurrence of at least 2 months. This led to nine mutations, each of them creating a defined cluster covering between 12 and 811 closely related genomes. We only kept mutation information of these nine subclusters in the respective metadata, which, together with the tree file of the phylogenetic time tree, was uploaded to a 'https://microreact.org/'-project, provided as *Supplementary file 1* and found under the following link: here.

## Mobile service data

T-Systems International GmbH collected and aggregated mobile service data via the Cell ID method, dividing a geographical area into the so-called traffic cells. Each cell is assigned to exactly one transmitter mast, with a spatial resolution from 500 m × 500 m up to 8 km × 8 km (depending on the transmitter mast network density). Cell phones always register to the closest traffic cell, which is recorded and stored in an origin-destination matrix (ODM). For population representation, the data was extrapolated with Deutsche Telekom's market share. Due to data privacy, the registration data is combined into movement streams between traffic cells, the status resolution is reduced to 1 hr (greater time intervals = less resolution), and individual traffic cells are grouped into districts. The degree of anonymization (k-value=30, data privacy regulation) removed movement streams with less than 30 participants, resulting in approximately 200 Mio trips in the ODM. SMA Development GmbH analyzed all movements between the single Thuringian districts, adding each Alpha sample's isolation time and location data (per subcluster). The movements were further divided by months and originating district (subcluster-affiliated vs. -unaffiliated), determining each district's monthly inbound mobility proportion from cluster-affiliated districts.

## Research in context

### Evidence before this study

We searched PubMed for studies about the use of mobile service data for surveillance written in English. For the broadest possible search, we included any publication covering mobile data and surveillance aspects, using the following search string: ('cellular data' OR 'cell phone data' OR 'mobility data' OR 'movement data' OR 'migration data' OR 'phone data') AND ('Surveillance' OR 'Monitoring' OR 'Survey' OR 'Pandemic' OR 'Disease' OR 'Epidemic' OR 'Outbreak'). Our search yielded 1285 publications published between 1966 and 2023. We manually screened all these publications but found no study that applied mobile service data for active, targeted surveillance. Across all studies, the general focus was on tracking contacts or analyzing movements to assess, for instance, the efficiency of non-pharmaceutical interventions or generate prediction models. Some studies suggested targeted surveillance based on their results, but it was not yet applied. Additionally, we used 'suite.ai' and 'chatGPT' (with BING-search access) to let them search for 'studies that utilize mobile service data to guide the sampling process for infectious disease surveillance'. While 'suite.ai' found two studies and 'chatGPT' found another ten studies and reviews, none covered the direct application of the mobility data in active surveillance.

### Added value of this study

This study highlights the value of combining mobile service data with fine-granular metadata for integrated genomic surveillance during the SARS-CoV-2 pandemic in a German federal state. We illustrated this strategy with the Omicron sublineage BQ.1.1 and how to guide the sampling processes toward areas where the new variant was expected to emerge. Additionally, we used mobile service data during the pandemic to assess our sampling coverage. Our study is the first to actively guide part of the genomic surveillance process during a pandemic.

### Implications of all the available evidence

Efficient molecular surveillance setups are crucial in managing outbreaks from the local to the global scale. Different data sources are investigated to increase this efficiency, addressing factors like the more efficient usage of scarce surveillance resources and the prediction of spread. Extending molecular surveillance with such data should improve the future management of pandemics and outbreaks.

## Declaration of generative AI and AI-assisted technologies in the writing process

During the preparation of this work, the author(s) used Grammarly in order to correct general English and improve readability. After using this tool/service, the author(s) reviewed and edited the content as needed and take(s) full responsibility for the content of the publication.

## Acknowledgements

Funding: This work was supported by grants from the Federal Ministry of Education and Research (project 'SARS-CoV-2Dx') (grant number 13N15745) and the Thüringer Aufbaubank (project 'Pandemie Analyse mittels Advanced Analytics Methoden') (grant number 2021 VF 0035). We acknowledge support by the German Research Foundation Projekt-Nr. 512648189 and the Open Access Publication Fund of the Thueringer Universitaets- und Landesbibliothek Jena.

## Additional information

### Funding

| Funder | Grant reference number | Author |
| --- | --- | --- |
| Bundesministerium für Bildung und Forschung | 13N15745 | Mathias W Pletz |
| Thüringer Aufbaubank | 2021 VF 0035 | Carolin Fleischmann-Struzek |
| German Research Foundation | 512648189 | Riccardo Spott |

The funders had no role in study design, data collection and interpretation, or the decision to submit the work for publication.

### Author contributions

Riccardo Spott, Writing – original draft, Writing – review and editing, Sample collection and preparation, sequencing, bioinformatic analysis, literature research; Mathias W Pletz, Funding acquisition, Writing – review and editing; Carolin Fleischmann-Struzek, Writing – review and editing, Funding acquisition; Aurelia Kimmig, Petra Dickmann, Martin Hölzer, Denise Kühnert, Writing – review and editing; Christiane Hadlich, Writing – review and editing, Mobile service data analysis; Matthias Hauert, Writing – review and editing, Mobile service data analysis; Mara Lohde, Sample collection and preparation, sequencing, bioinformatic analysis; Mateusz Jundzill, Database setup and maintenance; Mike Marquet, Writing – review and editing, Sample collection and preparation, sequencing, bioinformatic analysis; Ruben Schüchner, Writing – review and editing, Sample collection and preparation; Christian Brandt, Software, Supervision, Project administration, Writing – review and editing, Bioinformatic analysis

### Author ORCIDs

Riccardo Spott ⓘ https://orcid.org/0000-0002-2103-167X
Denise Kühnert ⓘ http://orcid.org/0000-0002-5657-018X

### Ethics

SARS-CoV-2 samples were collected and sequenced according to the German Corona-Surveillance Act issued by the German Health Ministry ("Verordnung zur molekulargenetischen Surveillance des Coronavirus SARS-CoV-2 (Coronavirus-Surveillanceverordnung - CorSurV) Vom 18. Januar 2021", BAnz AT 19.01.2021 V2, published 19th of Jan 2021). Ethics committee/IRB of the University Hospital Jena gave ethical approval for this work (No. UKJ_2018-1263_2-BO).Sampling was performed by medical institutions during routine clinical and epidemiological surveillance and samples were anonymized prior to analysis. Informed consent and consent to publication were not required in accordance with German health research regulations.

Reviewer #1 (Public review): https://doi.org/10.7554/eLife.94045.3.sa1
Reviewer #2 (Public review): https://doi.org/10.7554/eLife.94045.3.sa2
Author response https://doi.org/10.7554/eLife.94045.3.sa3

## Additional files

### Supplementary files

Supplementary file 1. 'microreact'-file summarizing all data presented in the microreact project. The project can be found at here.

Supplementary file 2. Description of supplementary method 'Phylogenetic time tree construction'.

Supplementary file 3. 'xz'-packed fasta-file, containing all SARS-CoV-2 Alpha lineage genomes used in the Nextstrain analysis.

Supplementary file 4. Metadata tsv-file containing the information for all SARS-CoV-2 Alpha lineage genomes used in the Nextstrain analysis.

Supplementary file 5. 'yaml'-file containing the build-instructions for the Nextstrain analysis.

Supplementary file 6. 'xz'-packed fasta-file containing the resulting, subsampled genomes of the Nextstrain analysis.

Supplementary file 7. Nextstrain 'tree.nwk'-file used to visualize the phylogenetic time tree. Contains 64,131 German (non-Thuringian) and 6298 Thuringian Alpha lineage genomes.

Supplementary file 8. Metadata tsv-file for the Alpha lineage genomes contained in the phylogenetic time tree.

Supplementary file 9. GeoData used in *Figure 1—source code 1*, *Figure 2—source code 1*, *Figure 3—source code 1*.

MDAR checklist

### Data availability

All genomic data (genomes and respective metadata), supportive data, and the R-scripts used to generate all figures are provided as supplementary files. Additionally, they are available at https://doi.org/10.17605/OSF.IO/N5QJ6. These include the mobile service data used in this study, which is available in processed and anonymized form. The original mobile service data can not be made public due to legal reasons/ownership. The genomic data is further available in the provided micro-react project (project file available as Supplementary File 1 and under https://doi.org/10.17605/OSF.IO/N5QJ6; https://microreact.org/project/ftR2GfjF6iXtSwbmN4ARTx-thuringianalpha-linclusters#76ir-complete-overview).

The following dataset was generated:

| Author(s) | Year | Dataset title | Dataset URL | Database and Identifier |
|---|---|---|---|---|
| Spott R | 2024 | Exploring the Spatial Distribution of Persistent SARS-CoV-2 Mutations - Supplementary files | https://doi.org/10.17605/OSF.IO/N5QJ6 | Open Science Framework, 10.17605/OSF.IO/N5QJ6 |

The following previously published dataset was used:

| Author(s) | Year | Dataset title | Dataset URL | Database and Identifier |
|---|---|---|---|---|
| Koch-Institut R | 2022 | SARS-CoV-2 Sequenzdaten aus Deutschland | https://doi.org/10.5281/zenodo.7212725 | Zenodo, 10.5281/zenodo.7212725 |

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
