## [Editor Report · eLife Assessment]

The authors analyze the relationship between human mobility and genomic data of SARS-CoV-2 using mobile phone mobility data and sequence data and present a **solid** proof of concept. This **useful** work was conducted on a fine spatial scale and provides suggestions on how mobility-derived surveillance could be conducted, although these results are mixed. The primary significance of this work is the strong use of large datasets that were highly granular. The authors provide a rigorous study, but with less clear predictive power of mobility to inform transmission patterns.

---

## [Referee Report · Reviewer #1 (Public review)]

Summary:

In this manuscript Spott et al. combine SARS-CoV-2 genomic data alongside granular mobility data to retrospectively evaluate the spread of SARS-CoV-2 alpha lineages throughout Germany and specifically Thuringia. They further prospectively identified districts with strong mobility links to the first district in which BQ.1.1 was observed to direct additional surveillance efforts to these districts. The additional surveillance effort resulted in the earlier identification of BQ.1.1 in districts with strong links to the district in which BQ.1.1 was first observed.

Strengths:

There are two important strengths of this work. The first, is the scale and detail in the data that has been generated an analyzed as part of this study. Specifically, the authors use 6,500 SARS-CoV-2 sequences and district level mobility data within Thuringia. I applaud the authors for making a subset of their analyses public e.g. on the associated micro react page.

Further, the main focus of the article is on the potential utility of mobility-directed surveillance sequence. While I may certainly be mistaken, I have not seen this proposed elsewhere, at least in the context of SARS-CoV-2. The authors were further able to test this concept in a real world setting during the emergence of BQ.1.1 and compare it to the "gold standard" of random sampling. This is a unique real-world evaluation of a novel surveillance sequencing strategy and there is considerable value in publishing this analysis. Given the increased focus on optimizing sampling strategies for genomic surveillance, this work provides a novel strategy and will hopefully motivate additional modeling and real-world implementations.

Weaknesses:

The article is quite strong and I find the analyses to generally be rigorous. Limitations of the analysis, particularly due to the fact that BQ.1.1 remained a low-prevalence variant, are adequately addressed. The results do not provide quantitative, definitive proof that mobility-guided sampling is an optimal strategy, but they also do not claim to nor do I think they need to to make an important contribution to the field.

---

## [Referee Report · Reviewer #2 (Public review)]

In the manuscript, the authors combine SARS-CoV-2 sequence data from a state in Germany and mobility data to help in understanding the movement of virus and the potential to help decide where to focus sequencing. The global expansion in sequencing capability is a key outcome of the public health response. However, there remains uncertainty how to maximise the insights the sequence data can give. Improved ability to predict the movement of emergent variants would be a useful public health outcome.

However, I remain unconvinced that changing surveillance strategies is necessarily sensible as it remains unclear what the ultimate benefit of variant hunting is. Decisions to adapt surveillance strategies should not be taken lightly as there are substantial benefits of maintaining a stable and as representative as possible, system over time. It's unclear what public health action would result of detecting a few more sequences of a variant. Once a variant has been identified (arguably anywhere in the world/region), we already have the necessary information to motivate the development of updated vaccines/monoclonals.

---

## [Author Response]

The following is the authors’ response to the original reviews.

Thank you for your assessment and constructive critique, which helped us to improve the manuscript and its clarity. Upon carefully reading through the comments, we noticed that, based on the Reviewer's questions, some of our answers were already available but “hidden” as supplementary data. Thus, we changed the following two figures and text accordingly to showcase our results to the reader better:

A) To highlight how mobile service data can indicate the spread of highly prevalent variants, we added a high-prevalence subcluster to Figure 2 (previously shown in Supplementary Figures S4 and S5) and, in exchange, moved one low-prevalence subcluster from Figure 2 back into the supplement. The figure is now showing a low and a high prevalent subcluster instead of two low prevalent subclusters.

B) Based on Reviewer 1’s question about where samples were taken in regards to the mobility data from the community of the first identification (negative controls), we now highlight all the mobility data that was available to us in Figure 3 (as triangles) instead of just a few top mobility hits for both - mobility guided and random surveillance (serving as a negative control for the former). This way, we think, it is clearer how random sampling was also performed in some regions where mobility was coming from the community of origin (as asked by Reviewer 1) - the detailed trips and sampling are now part of the supplement for data transparency reasons. We also noticed a typo in the GPS coordinates, aligning one of the arrows falsely, which is corrected in the improved Figure 3.

We have also included the R-Scripts used to generate all the figures in the manuscript in an OSF repository (we updated the “Data sharing statement”). We also updated Figure 1 slightly and extended the supplemental material. The remaining comments to reviewers are addressed point-by-point below.

**Reviewer 1 (Public Review):**
In "1 Exploring the Spatial Distribution of Persistent SARS-CoV-2 Mutations -Leveraging mobility data for targeted sampling" Spott et al. combine SARS-CoV-2 genomic data alongside granular mobility data to retrospectively evaluate the spread of SARS-CoV-2 alpha lineages throughout Germany and specifically Thuringia. They further prospectively identified districts with strong mobility links to the first district in which BQ.1.1 was observed to direct additional surveillance efforts to these districts. The additional surveillance effort resulted in the earlier identification of BQ.1.1 in districts with strong links to the district in which BQ.1.1 was first observed.

Thank you for taking the time to review our work.

(1) It seems the mobility-guided increased surveillance included only districts with significant mobility links to the origin district and did not include any "control" districts (those without strong mobility links). As such, you can only conclude that increasing sampling depth increased the rate of detection for BQ.1.1., not necessarily that doing so in a mobility-guided fashion provided an additional benefit. I absolutely understand the challenges of doing this in a real-world setting and think that the work remains valuable even with this limitation, but I would like the lack of control districts to be more explicitly discussed.

Thank you for the critical assessment of our work. We agree that a control is essential for interpreting the results. In our case, randomized surveillance (“the gold standard”) served as a control with a total sampling depth seven times higher than the mobility-guided sampling. To better reflect the sampling in regards to the available mobility data, we revisited Figure 3 and added all the mobility information from the origin that was available to us. We also added this information to the random surveillance to provide a clearer picture to the reader. This now clearly shows how randomized surveillance covered communities with varying degrees of incoming mobility from the community of first occurrences, thereby underlining its role as a negative control. We updated the manuscript to reflect these changes and included the October 2020 and June 2021 mobility datasets in Supplementary Table S6. We agree that the sampling depth increases the detection, which is the point of guided sampling to increase sampling, specifically in areas where mobility points towards a possible spread. In regards to the negative control: Random surveillance (not Mobility-guided) in October covered 40 samples in the northwest region of Thuringia (Mobility-guided covered 19 samples). Thus, random surveillance also contained 31 out of 132 samples with a mobility link towards the first occurrence of BQ1.1 but with varying amounts of mobility (low to high).

We added this information to the main text:

Line 270 to 293:

Following its first Thuringian identification, we utilized the latest available dataset of the past two years of mobile service data (October 2020 and June 2021) to investigate the residential movements for the community of first detection. Considering the highest incoming mobility from both datasets, we identified 18 communities with high (> 10,000), 34 with medium (2,001-10,000), and 82 with low (30-2,000) number of incoming one-way trips from the originating community (purple triangles in Figure 3a). As a result, we specifically requested all the available samples from the eight communities with the highest incoming mobility. Still, we were restricted to the submission of third parties over whom we had no influence. This led to the inclusion of the following eight communities with the most residential movement from the originating community: four in central and three in NW of Thuringia, one in NW-neighboring state Saxony-Anhalt. The samples requested from central Thuringia were also due to their geographic arrangement as a “belt” in central Thuringia, linking three major cities (see Supplementary Figure S1). Subsequently, we collected 19 additional samples (isolated between the 17th and 25th of October 2022; see “Guided Sampling” for October 2022, Figure 3a) besides the randomized sampling strategy. Thus, the sampling depth was increased in communities with high incoming mobility from the first origin.

As part of the general Thuringian surveillance, we collected 132 samples for October (covering dates between the 5th and 31st) and 69 samples in November (covering dates between the 1st and 25th; see Figure 3b and c). Randomized sampling was not influenced or adjusted based on the mobility-guided sample collection. Thus, it also contains samples from communities with a mobility link towards the first occurrence of BQ.1.1, as they were part of the regular random collection (see gray triangles in Figure 3b). A complete overview of all samples is provided in Supplementary Table S5. The mobility datasets from October 2020 and June 2021 for all sampled communities are provided in Supplementary Table S6.

Line 305 to 313:

Among the 19 samples specifically collected based on mobile service data, we identified one additional sample of the specific Omicron sublineage BQ.1.1 in a community with high incoming mobility (n = 14, number of trips = 37,499) with a distance of approximately 16 km between both towns. Our randomly sampled routine surveillance strategy did not detect another sample during the same period. This was despite a seven times higher overall sample rate, which included 31 samples from communities with an identified incoming mobility from the community of the first occurrence (October 2022, Figure 3b). Only in the one-month follow-up were four other samples identified across Thuringia through routine surveillance (November 2022, Figure 3c).

Line 325 to 333:

In summary, increasing the sampling depth in the suspected regions successfully identified the specified lineage using only a fraction of the samples from the randomized sampling. Conversely, randomized surveillance, the “gold standard” acting as our negative control, did not identify additional samples with similar sampling depths in regions with no or low incoming mobility or even in high mobility regions with less sampling depth. Implementing such an approach effectively under pandemic conditions poses difficult challenges due to the fluctuating sampling sizes. Although the finding of the sample may have been coincidental, our proof of concept demonstrated how we can leverage the potential of mobile service data for targeted surveillance sampling.

(2) Line 313: While this work has reliably shown that the spread of Alpha was slower in Thuringia, I don't think there have been sufficient analyses to conclude that this is due to the lack of transportation hubs. My understanding is that only mobility within Thuringia has been evaluated here and not between Thuringia and other parts of Germany.

Thank you for pointing this out. We noticed that the original sentence lacked the necessary clarity. The statement in line 313 was based on the observation that Alpha first occurred in federal states with major transport hubs, such as international airports and ports, which Thuringia lacks, as demonstrated in the Microreact dataset. For clarification, we adjusted the sentence as follows:

Line 340 and following:

A plausible explanation for the delayed spread of the Alpha lineage in Thuringia is the lack of major transport hubs, as Alpha first occurred in federal states with such hubs. Previous studies have already highlighted the impact of major transportation hubs in the spread of Sars-CoV-2.

(3) Line 333 (and elsewhere): I'm not convinced, based on the results presented in Figure 2, that the authors have reliably identified a sampling bias here. This is only true if you assume (as in line 235) that the variant was in these districts, but that hasn't actually been demonstrated here. While I recognize that for high-prevalence variants, there is a strong correlation between inflow and variant prevalence, low-prevalence variants by definition spread less and may genuinely be missing from some districts. To support this conclusion that they identified a bias, I'd like to see some type of statistical model that is based e.g. on the number of sequences, prevalence of a given variant in other districts, etc. Alternatively, the language can be softened ("putative sampling bias").

Thank you for addressing this legitimate point of criticism in our interpretation. Due to the retrospective nature of the analysis and the fact that we found no additional samples of the clusters after the specified timeframes, we were limited to the samples in our dataset. Therefore, it is impossible to demonstrate if a variant was present in the relevant districts afterward. We agree that the variant’s low prevalence means they may genuinely not have spread to some districts. For clarification, we added the following statements and changed the wording accordingly:

Additional statement in line 248:

However, due to their low prevalence, it is also possible that these subclusters have not spread to the indicated districts.

Adjusted wording in line 361:

We exemplified this approach with the Alpha lineage, where mobile service data indicated a putative sampling bias and partially predicted the spread of our Thuringian subclusters.

**Recommendations:**
(1) I applaud the use of the microreact page to make the data public, however, I don't see any reference to a GitHub or Zenodo repository with the analysis code. The NextStrain code is certainly appreciated but there is presumably additional code used to identify the clusters, generate figures, etc. I generally prefer this code be made public and it is recommended by eLife.

Thank you for your appreciation. We have now included the R-scripts in the manuscript’s OSF repository. These were used to create the figures in the manuscript and supplement utilizing the supplementary tables 1-6, which are also stored in the repository. To clearly communicate which data is provided, we changed lines 513 and 514 of the “Data sharing statement” as follows:

Line 513 and following:

Supplementary tables and the R-scripts used to generate all figures are also provided in the repository under https://osf.io/n5qj6/. These include the mobile service data used in this study, which is available in processed and anonymized form.

The subcluster identification was performed manually. By adding each sample's mutation profile to the Microreact metadata file, we visually screened the phylogenetic time tree for all non-Alpha specific mutations present in at least 20 Thuringian genomes. We then applied the criteria described in the Methods section to identify the nine Alpha subclusters. For clarification, we changed line 436:

Line 436:

We then manually screened for mutations present in at least 20 genomes with a small phylogenetic distance and a time occurrence of at least two months.

**Reviewer 2 (Public Review):**
In the manuscript, the authors combine SARS-CoV-2 sequence data from a state in Germany and mobility data to help in understanding the movement of the virus and the potential to help decide where to focus sequencing. The global expansion in sequencing capability is a key outcome of the public health response. However, there remains uncertainty about how to maximise the insights the sequence data can give. Improved ability to predict the movement of emergent variants would be a useful public health outcome. Also knowing where to focus sequencing to maximising insights is also key. The presented case study from one State in Germany is therefore a useful addition to the literature. Nevertheless, I have a few comments.

Thank you for taking the time to review our work.

(1) One of the key goals of the paper is to explore whether mobile phone data can help predict the spread of lineages. However, it appears unclear whether this was actually addressed in the analyses. To do this, the authors could hold out data from a period of time, and see whether they can predict where the variants end up being found.

Based on your feedback, we noticed that the results of the other seven clusters presented in the supplement were not appropriately highlighted, causing them to be overlooked. We indeed demonstrated that predicting viral spread based on mobility data is possible, as shown for the high-prevalence subcluster 7 (Cluster “ORF1b:A520V”, 811 samples). This was briefly mentioned in lines 240-242, but the cluster was only shown in Supplementary Figures S4 and S5. Instead, we focused more on the putative sampling bias that the mobility for low-prevalence subclusters could indicate as an interesting use case of mobility data. This addresses a concrete problem of every surveillance: successfully identifying low-prevalence targets. However, based on your feedback, we revisited Figure 2, adding the plots of the high-prevalence subcluster: “ORF1b:A520V” from Supplementary Figures S4 and S5 while moving the low-prevalence subcluster “S:N185D” from Figure 2 into the Supplementary Figures S4 and S5. Additionally, we changed line 229 to highlight this result properly.

line 229 and following:

The mobile service data-based prediction of a subcluster’s spread aligned well with the subsequent regional coverage of fast-spreading, highly prevalent subclusters, such as subcluster 7, which covered 811 samples (see Figure 2). In contrast, the predicted spread for the low-prevalence subclusters did not correspond well with the actual occurrence.

(2) The abstract presents the mobility-guided sampling as a success, however, the results provide a much more mixed result. Ultimately, it's unclear what having this strategy really achieved. In a quickly moving pandemic, it is unclear what hunting for extra sequences of a specific, already identified, variant really does. I'm not sure what public health action would result, especially given the variant has already been identified.

Thank you for your critical assessment of the presented results and their interpretation.

Here, we aimed to provide an alternative to the standard randomized surveillance strategy. Through mobility-guided sampling, we sought to increase identification chances while necessitating fewer samples and decreasing costs, ultimately enhancing surveillance efficiency. The Omicron-lineage BQ.1.1 was the perfect example to prove this concept under actual pandemic conditions. Yet, the strategy is not limited to low-prevalence sublineages but can be applied to virtually any surveillance case. However, from your question, we recognize that this conclusion was unclear from the text. Therefore, we adapted the conclusion to better communicate the real implications of our proof of concept. Additionally, we altered line 42 in the abstract for clarification.

However, we did not assess the benefits of surveillance itself, as the German Robert Koch Institute (RKI) already had outlined its importance for tracking different viral variants. This tracking served several reasons, like monitoring vaccine escapism, mutational progress, and assessing available antibodies for treatment.

Line 42:

The latter concept was successfully implemented as a proof-of-concept for a mobility-guided sampling strategy in response to the surveillance of Omicron sublineage BQ.1.1.

Line 364 to 374:

Another approach is actively guiding the sampling process through mobile service data, which we demonstrated with our proof of principle focusing on the Omicron-lineage BQ.1.1 as a real-life example. This approach could allow for a flexible allocation of surveillance resources, enabling adaptation to specific circumstances and increasing sampling depth in regions where a variant is anticipated. By incorporating guided sampling, much fewer resources may be needed for unguided or random sampling, thereby reducing overall surveillance costs.

Additionally, while this approach is particularly useful for identifying low-prevalence variants, it is not limited to such variants. Still, it can provide a guided, more cost-efficient, low-sampling alternative to general randomized surveillance that can also be applied to other viruses or lineages.

(3) Relatedly, it is unclear to me whether simply relying on spatial distance would not be an alternative simpler approach than mobile phone data. From Figure 2, it seems clear that a simple proximity matrix would work well at reconstructing viral flow. The authors could compare the correlation of spatial, spatial proximity, and CDR data.

Thank you for pointing this out. While proximity data might appear to be an obvious choice, it has significant limitations compared to mobility data, especially in the context of our study. Proximity data assumes that spatial distance alone can accurately represent movement patterns, which would only be true in a normally distributed traffic network. Geographic features such as mountains, cities, and highways affect traffic flows, leading to variability over distance and time, which are beyond the scope of spatial proximity but efficiently captured by mobility data. In Figure 2, we presented a simplified view of the mobility data. Hence, proximity and mobility data appear to provide the same insights. However, as shown in the updated Figure 3, a detailed overview of the available mobility data reveals obvious and non-obvious spatial connections that proximity data can not capture. Incorporating such a level of detail in Figure 2 would have cluttered the figure and reduced its clarity (e.g., adding triangles for each Thuringian community).

While a comparison between proximity data and mobility data would indeed be informative, it is beyond the scope of our current study, as our primary focus was to examine the useability of mobility data in explaining our subcluster’s spread in the first place. However, we agree it would be a valuable direction for future research. We summarized our thoughts from above in the following additional sentence:

Line 374:

Pre-generated mobility networks automatically tailored to each state's unique infrastructure and population dynamics could provide better-targeted sampling guidance rather than simple geographical proximity.

**Recommendations:**
(1) Line 128: What do these percentages mean - the proportion of States with at least one Alpha variant? Please clarify.

We clarified the values at their first appearance in the text:

Line 127:

By March, Alpha had spread to nearly all states and districts (districts are similar to counties or provinces) in Germany (Median: 76·47 % Alpha samples among a federal states total sequenced samples compared to 36·03 % in February, excluding Thuringia) and Thuringia (Median: 85·29 %, up from 50·00 % in February).

(2) Line 134: It's a little strange to compare the dynamics of a state with that of the whole country. For it lagged as compared to all other States?Line 134: “In summary, the spread of the Alpha lineage in Thuringia lagged roughly two weeks behind the general spread in the rest of Germany but showed similar proportions.”

Thank you for the feedback. The statement refers to the comparison of Alpha-lineage proportions across federal states, excluding Thuringia, in lines 118 to 130. To simplify, we collectively referred to these federal states as “Germany” in the text. However, we recognize that this formulation is misleading, so we adjusted line 135 for clarification:

Line 135:

In summary, the spread of the Alpha lineage in Thuringia lagged roughly two weeks behind the general spread of other German federal states but showed similar proportions.